# Quantum behavior of the Duffing oscillator at the dissipative phase transition

Qi-Ming Chen [1,2] ✉, Michael Fischer[1,2], Yuki Nojiri[1,2], Michael Renger[1,2], Edwar Xie[1,2], Matti Partanen [1,4], Stefan Pogorzalek[1,2,5], Kirill G. Fedorov[1,2], Achim Marx[1], Frank Deppe [1,2,3,5] ✉ & Rudolf Gross [1,2,3] ✉

The non-deterministic behavior of the Duffing oscillator is classically attributed to the coexistence of two steady states in a double-well potential. However, this interpretation fails in the quantum-mechanical perspective which predicts a single unique steady state. Here, we measure the non-equilibrium dynamics of a superconducting Duffing oscillator and experimentally reconcile the classical and quantum descriptions as indicated by the Liouvillian spectral theory. We demonstrate that the two classically regarded steady states are in fact quantum metastable states. They have a remarkably long lifetime but must eventually relax into the single unique steady state allowed by quantum mechanics. By engineering their lifetime, we observe a first-order dissipative phase transition and reveal the two distinct phases by quantum state tomography. Our results reveal a smooth quantum state evolution behind a sudden dissipative phase transition and form an essential step towards understanding the intriguing phenomena in driven-dissipative systems.

The Duffing oscillator is a simple but prototypical model in nonlinear physics, which describes a forced oscillation with cubic (Kerr) nonlinearity and linear viscous damping[1]. In a certain parameter regime, classical mechanics predicts a double-well potential that allows two steady states (SSs) at the same parameter setting[2]. It gives rise to a hysteretic behavior where two different amplitudes of the forced oscillation are possible. Depending on whether the system is initially at rest or in strong oscillation, the oscillator spontaneously chooses one of the amplitudes when adiabatically tuning the parameters into the hysteretic regime. Thermal fluctuations may induce unpredictable jumps between the two potential wells and lead to the bistability of the oscillation amplitude[3]. This classical behavior of the Duffing oscillator has been observed in a considerable number of experiments, for example, in superconducting quantum circuits[4–6]. The underlying double-well potential model has been used to explain a variety of physical processes, such as optical bistability[7,8], parametric amplification[9,10], and self-oscillation[11,12]. However, it has been revealed by Drummond and Walls already in the 1980s that a fully-quantum treatment of the Duffing oscillator yields a single unique SS over the entire parameter space, such that it does not exhibit bistability or hysteresis[13]. These two perspectives indicate fundamentally different behaviors of the Duffing oscillator. However, the seeming two classical SSs are still observed even in a typical quantum experiment setup[5,6]. Recently, signatures of dissipative phase transition (DPT) have been observed in the scattering coefficient[14,15], decay rate[15,16], and second-order correlation function[17] of the Duffing oscillator, which indicate a prominent role of the quantum fluctuation in the SS. These experiments are performed around a fixed parameter setting in a continuous-wave measurement setup.

Here, we use an in-situ tunable superconducting nonlinear resonator to simulate the non-equilibrium quantum dynamics of the Duffing oscillator. Besides the wide tunability range of sample

[1]Walther-Meißner-Institut, Bayerische Akademie der Wissenschaften, 85748 Garching, Germany. [2]Physik-Department, Technische Universität München, 85748 Garching, Germany. [3]Munich Center for Quantum Science and Technology (MCQST), 80799 Munich, Germany. [4]Present address: IQM, Keilaranta 19, FI-02150 Espoo, Finland. [5]Present address: IQM, Nymphenburger Str. 86, 80636 Munich, Germany. ✉e-mail: qiming.chen@wmi.badw.de; frank.deppe@wmi.badw.de; rudolf.gross@wmi.badw.de

parameters in one device, the pulsed heterodyne measurement distinguishes our experiment from the experiments already reported in the literature. Our experimental setup allows for a proper control of the initial state at different parameter settings as well as a high time resolution readout. Our experimental results settle the seeming controversy between the classical and quantum properties of the Duffing oscillator and provide support to the recent results of the Liouvillian spectral theory[18–21]. We demonstrate that the two classical SSs are in fact quantum metastable states (MSs), which emerge when the low-lying eigenvalues of the Liouvillian superoperator are separated from the rest of its spectrum[18]. Different from the classical MSs that are unstable SS solutions of the equation of motion, quantum MSs have a lifetime much longer than any other timescale in the system but are not the exact SS solutions of the Schrödinger equation. A remarkable feature occurs when the system approaches the thermodynamic limit, where the MSs gain an increasingly long lifetime when approaching a critical point but, suddenly, cannot be properly defined at the exact point[19,20]. This non-analytical phenomenon is identified as a first-order DPT, which originates from the interplay between a coherent drive and an incoherent dissipation in a nonlinear driven-dissipative system.

## Results

### Liouvillian spectral analysis

The non-equilibrium quantum dynamics of the Duffing oscillator is described by the master equation in the Born-Markov approximation: $\partial_t \rho(t) = \mathcal{L}\rho(t)$, where the Liouvillian superoperator, $\mathcal{L}$, consists of the Kerr-oscillator Hamiltonian with coherent drive, $H/\hbar = \Delta a^\dagger a + U a^\dagger a^\dagger a a + \xi(a + a^\dagger)$, and the Lindblad superoperator, $(\gamma/2)\mathcal{D}[a]$. Besides, $a$ ($a^\dagger$) is the annihilation (creation) operator of the oscillating mode, $\Delta$ the detuning between the resonant frequency and the drive, $U$ the Kerr nonlinearity, and $\xi$ the driving strength. We define $\gamma$ as the total energy dissipation rate, which is approximately $3.85\,\mu s^{-1}$ in the measured frequency range, and we neglect the relatively weak dephasing effect (See Supplementary Fig. 4). When restricting our discussion to finite dimensions, the Liouvillian superoperator can be decomposed into Jordan blocks that lead to the formal solution: $\rho(t) = \sum_n \exp(\lambda_n t)\left(\sum_m c_{n,m} r_{n,m}\right)$ with $c_{n,m} = \text{tr}\left[l_{n,m}\rho(0)\right]$. Here, $l_{n,m}$ and $r_{n,m}$ are the left and right eigenmatrices of $\mathcal{L}$, which correspond to the $n$th eigenvalue with geometric multiplicity $m$. For convenience, we define $\delta_n = -\text{Re}(\lambda_n)$ and sort the eigenvalues according to $\delta_n < \delta_{n+1}$. Under quite general conditions, there exists a single unique SS solution such that $\delta_0 = 0$, $\delta_1 > 0$[22]. Thus, the smallest nonzero eigenvalue, forming the Liouvillian gap $\delta_1$, determines the timescale the system requires to relax into the SS, and thereby results in a general exponential decay of an observable. However, if the Liouvillian gap is well separated from the rest of its spectrum, $\delta_1 \ll \delta_2$, the system may quickly relax onto the metastable manifold spanned by $r_{0,1}$ and $\{r_{1,m}\}$ within a timescale of $1/\delta_2$, and stays almost invariant for a relatively large timescale, $1/\delta_1$, before starting a second relaxation into the unique SS[18]. The Liouvillian gap may even close at a single critical point in the thermodynamic limit, where the eigenvalue zero becomes degenerate at the exact point. The SS thus must undergo a sudden change on the two sides of the critical point and result in a first-order DPT[20].

### In-situ tunable Duffing simulator

The system studied in our experiment is realized by embedding a weakly asymmetric superconducting quantum interference device (SQUID) in the middle of a coplanar waveguide resonator. By driving it with a coherent microwave field, we implement a Duffing oscillator in superconducting quantum circuits with tunable frequency and Kerr nonlinearity[23,24], as shown in Fig. 1a. In our experiment, the resonant frequency, $\omega_A/2\pi$, is varied between 6.80 GHz and 7.15 GHz, corresponding to a tunable range of the nonlinearity from $U/2\pi = -295$ kHz to $-58$ kHz. We modulate the radio-frequency (RF) drive by three

different pulse shapes, which balance the depths of the two potential wells and prepare the system in one of the two wells or in the SS at the initial time (See Methods Section). Then, we switch the driving strength to $\xi$, and trigger a short measurement of the transmitted or reflected microwave signal after a time delay of $\tau$. In each repetition, the measurement lasts for only 16 ns to capture the transient non-equilibrium dynamics of the system. We repeat this procedure for more than one million times and concatenate the results in a long trace for extracting the quadrature histogram of the transient outgoing field (See Methods Section). Eventually, we obtain the quasi-distribution functions of the intra-resonator field for different initial states and also different control parameters, $\Delta$, $\xi$, and $\tau$.

### Quantum features behind classical hysteresis

In our experiment, we first tune the nonlinearity to $U/2\pi = -132$ kHz and drive the system with a varying strength, $\xi$, and detuning, $\Delta$. The measurement is delayed by $\tau = 3.25\,\mu s$, which is more than 10 times longer than the free-relaxation time of the resonator, $1/\gamma$. When the system is initially prepared in one of the two potential wells, the absolute mean field, $|\langle a \rangle|$, and the photon number, $\langle a^\dagger a \rangle$, show an abrupt change at either of the two boundaries of the classical hysteretic regime, as shown in Fig. 1b. Within this regime, the measured values are also different for the same parameter setting, which correspond to the two possible oscillation amplitudes of the Duffing oscillator, i.e., the two classical SSs in a double-well potential. However, the transition occurs inside the regime when applying a constant driving field, which corresponds to an infinitely large measurement delay in either of the two former cases. Classically, this is explained by the presence of thermal fluctuations that induce random jumps between the two potential wells and wash out the dependence on the initial-state for large $\tau$. However, this interpretation fails in our experimental situation where the thermal noise at the 30 mK base temperature is much smaller than half a photon, i.e., the vacuum quantum fluctuation, and thus not likely to cause a noticeable transition between the two potential wells.

Indeed, quantum fluctuations play a significant role in the hysteretic regime, as shown in Fig. 1c for a fixed detuning frequency, $\Delta/2\pi = 2.36$ MHz. A clear dip in the $|\langle a \rangle|$ curve is observed during the transition process, which is predicted as a result of out-of-phase quantum fluctuations in the unique SS[13–15]. By comparison, $\langle a^\dagger a \rangle$ is a monotonic function of $\xi$ since it is insensitive to the phase of quantum fluctuations. Moreover, the second-order correlation function, $g^{(2)}(0)$, is strongly peaked around the transition point and approaches unity for large $\xi$. This is a typical signature of a first-order DPT, resulting from a drastic change of the SSs on the two sides of a single critical point[17]. Importantly, we observe these quantum-mechanical signatures in company with the classical hysteretic behavior, indicating that the system may not have reached the real SS even for $\tau > 10/\gamma$. This new perspective suggests that the two classically regarded SSs may be interpreted as MSs with a remarkably long lifetime in the hysteretic regime[19–21]. The specific MS, in which the systems is staying, is determined by the distance between the initial state and the two MSs[18]. This leads to the seemingly classical behavior of hysteresis in Fig. 1b.

### Two-stage relaxation of the MSs

According to the theory of quantum metastability, the MSs exist only in the time window $1/\delta_2 \ll \tau \lesssim 1/\delta_1$ and should eventually relax into the single unique SS for $\tau \gg 1/\delta_1$[18]. To verify this prediction, we then fix the detuning frequency at $\Delta/2\pi = 2.01$ MHz and measure the complex reflection coefficient, $S_{22}$, for varying driving strength, $\xi$, and measurement delay, $\tau$. In these measurements, the nonlinearity is fixed at $U/2\pi = -71$ kHz. In Fig. 2a, we plot the reflection coefficients, corresponding to the two MSs, in the complex plane. The two MS branches form a closed loop for each fixed $\tau$, manifesting the classical signature of hysteresis around $\xi/2\pi = 1.51$ MHz. However, different from the

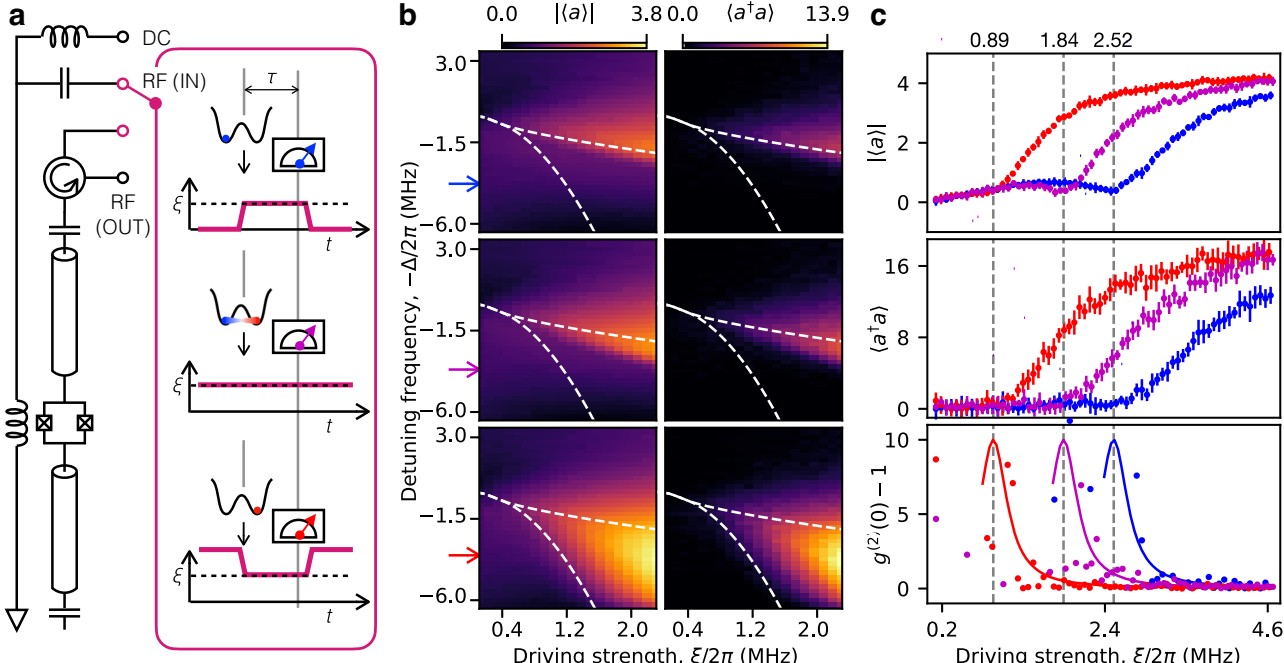

**Fig. 1 | Hysteresis and its quantum features. a** Schematic of the experimental setup for pulsed heterodyne measurement. The Duffing oscillator is initially prepared in one of the two potential wells (blue and red) or in the steady state (purple). Then, we switch the driving strength to $\xi$, and trigger a short measurement after a waiting time of $\tau$. The direct-current (DC) port is used to control the nonlinearity of the resonator, and we drive the resonator through one of the two radio-frequency (RF) ports for transmission- or reflection-type measurements (See Methods section). **b** The absolute mean field, $|\langle a \rangle|$, and photon number, $\langle a^\dagger a \rangle$, measured for $\omega_N/2\pi = 7.00$ GHz show a clear dependence on different initial states in the classical hysteretic regime, which is enclosed by the dashed curves calculated without any

fitting parameter (See Supplementary Note 1). A drastic change happens at either of the two boundaries if the system is initially prepared in one potential well. **c** At a fixed detuning frequency marked by the arrows in (**b**), the $|\langle a \rangle|$ vs. $\xi$ curves show a dip around the transition point (vertical dashed lines), while $\langle a^\dagger a \rangle$ is a monotonic function of $\xi$. The error bars represent the standard deviation over 8 independent experiments. The second-order correlation function $g^{(2)}(0)$ is strongly peaked around the transition point, which decays towards unity for large $\xi$. The solid curves are Lorentzian functions serving as guides to the eyes. Source data are provided as a Source Data file.

classical interpretation the loop exists within a decreasing parameter range when increasing $\tau$. It is expected to close for $\tau > 55\ \mu$s, where the two MS branches converge to the single unique SS allowed by quantum mechanics (See Supplementary Fig. 8). This observation provides a clear evidence for the quantum description of the Duffing oscillator. It indicates that the hysteresis observed in the classical hysteretic regime is the measurement outcome on two different MSs, while the system should eventually converge to a single unique SS in the long-time limit.

To quantify this convergence, we calculate the loop area, $A$, for different $\tau$, as shown in Fig. 2b. Fitting the data by an exponential decay, $A \propto \exp(-\eta\tau)$, we obtain two distinctively different decay rates $\eta_1 = 0.74\ \mu$s$^{-1}$ and $\eta_2 = 0.04\ \mu$s$^{-1}$ at small and large $\tau$, respectively. This two-stage relaxation process is qualitatively different from the classical prediction[25], but can be well understood from the Liouvillian spectrum[16,26]. Figure 2c shows the fitted Liouvillian gap, $\delta_1$, as a function of the driving strength, $\xi$. The gap is approximately $3.79\ \mu$s$^{-1}$ for both small and large $\xi$, what agrees well with the free energy decay rate $\gamma$. However, $\delta_1$ decreases by more than two orders of magnitude when approaching the critical point, $\xi^*$, and reaches a minimum value of $0.02\ \mu$s$^{-1}$ at $\xi^*$. This observation indicates a critical slowing down of the system dynamics around the critical point. This is another signature of a first-order DPT[15]. For a sufficiently small $\tau$, the decay rate of the loop area, $\eta_1$, is determined by the average value of the Liouvillian gap over the hysteretic regime, that is, $1.22\ \mu$s$^{-1}$. However, for $\tau \to \infty$ the decay rate $\eta_2$ is dominated by the minimum gap. In the time window between the two extreme cases, the decay rate decreases monotonically with $\tau$ and connects the two extremes. This is in quantitative agreement with the observed two-stage relaxation rates in Fig. 2b.

## The first-order DPT

It is then natural to ask whether the Liouvillian gap can be closed at a particular parameter setting, where the system dynamics becomes infinitely slow and the two MSs become also SSs. However, this perception is in conflict with the uniqueness of the SS solution for the Duffing oscillator[13]. Nevertheless, multiple SSs can exist in the driven-dissipative Bose-Hubbard model, where an infinite number of Duffing oscillators are coupled to each other and form a lattice. A bridge between the mean field description of an $N$-site Bose-Hubbard lattice and a single Duffing oscillator may be constructed by rescaling the nonlinearity and driving strength of the later as $U \to U_0/N$ and $\xi \to \sqrt{N}\xi_0$[19]. A thermodynamic limit of the Duffing oscillator is defined as $N \to \infty$, where the Liouvillian gap is closed at a rescaled critical point, $\xi_0^*$, and results in a first-order DPT.

In our experiment, we in situ tune the scaling factor from approximately $N = 2$ to 11, and measure the average photon number of the SS for varying driving strength. Here, we define $U_0 \equiv -\gamma$ for $N = 1$ and fix the detuning at $\Delta = 3\gamma$ without loss of generality[19]. As shown in Fig. 3a, the transition happens at the same rescaled critical driving strength, $\xi_0^*/2\pi = 0.64$ MHz, for different $N$, and the photon density also saturates at a similar value of around $\langle a^\dagger a \rangle/N = 2.5$. However, the transition process becomes sharper with increasing $N$. Fitting the data by a linear function, we obtain $\partial(\langle a^\dagger a \rangle/N)/\partial(\xi_0/2\pi) = 3.8$, 5.8, and 8.5 MHz$^{-1}$ for $N = 6.1$, 7.9, and 10.5, respectively, as shown in Fig. 3b. This observed tendency indicates a sudden change of the photon density on the two sides of $\xi_0^*$ in the thermodynamic limit, known as the first-order DPT[20]. It implies a closed Liouvillian gap, equivalently, a diverging lifetime of the MSs, when approaching the critical point. This is consistent with the observed Liouvillian gap in Fig. 2c.

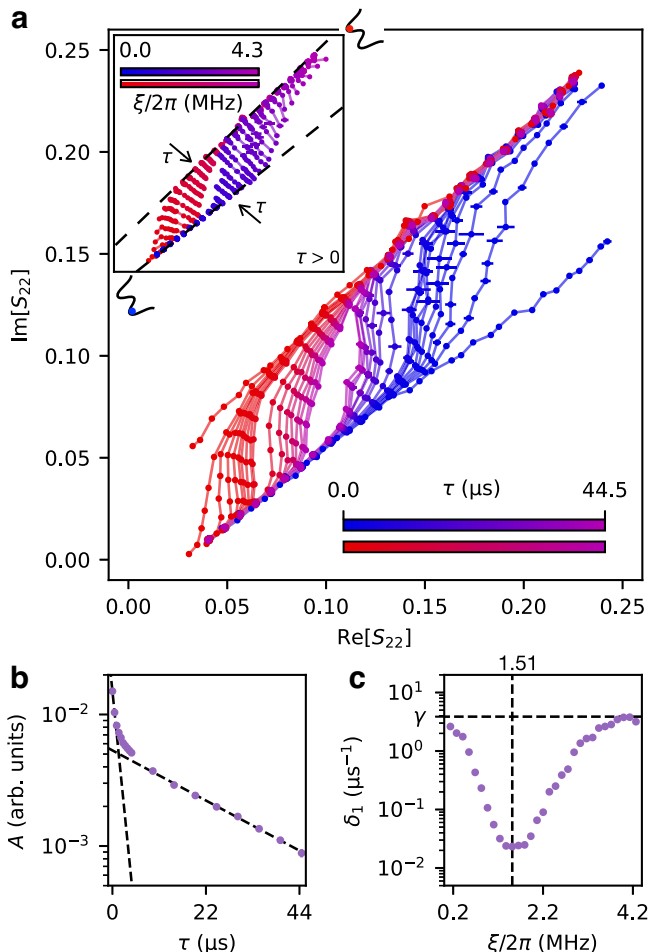

**a**

**b**

**c**

**Fig. 2 | Two-stage relaxation towards the single unique steady state. a** The reflection coefficients, $S_{22}$, corresponding to the two metastable branches (blue and red) form a closed loop, which converge to the unique steady-state solution with $\tau$. The inset shows the convergence of the metastable branches at each fixed $\xi$. **b** The loop area, $A$, decays with $\tau$ and shows two distinct decay rates. The dashed lines show the exponential fits, $A \propto \exp(-\eta\tau)$, of the decay rate at small and large $\tau$, with fitted values $\eta_1 = 0.74\,\mu s^{-1}$ and $\eta_2 = 0.04\,\mu s^{-1}$, respectively. **(c)** The Liouvillian gap, $\delta_1$, is approximately equal to the total energy dissipation rate, $\gamma$, at a sufficiently small or large $\xi$ (dashed). However, it decreases by more than two orders of magnitude when approaching the critical point, $\xi/2\pi = 1.51$ MHz, and achieves a minimum value of $0.02\,\mu s^{-1}$ at $\xi$ (See Supplementary Fig. 9). In all panels, the resonant frequency is fixed at $\omega_A/2\pi = 7.10$ GHz. The error bars represent the standard deviation over 16 independent experiments, which are smaller than the size of the dots in (b) and (c). Source data are provided as a Source Data file.

## Quantum state tomography during phase transition

To understand the underlying physical process of DPT, we reconstruct the Wigner quasi-distribution function of the intra-resonator field according to the first two orders of signal moments, $\langle a \rangle$, $\langle a^\dagger a \rangle$, and $\langle a^2 \rangle$ (See Supplementary Fig. 10). Here, we operate the SQUID close to its sweet spot where the dephasing rate is sufficiently small, as indicated by the good agreement between theory and experiment in Fig. 3a. As shown in Fig. 4, the reconstructed SS is approximately an either coherent or squeezed state in one of the two phases[27], where the field mean coincides with the two classical SS solutions[19]. In each individual phase, the SS remains almost invariant with respect to the rescaled driving strength. However, the system undergoes a drastic change in a relatively small range around the critical point, $0.57\,\text{MHz} \le \xi_0/2\pi \le 0.71$ MHz. This results in the rapid photon number transition in Fig. 3a. In this regime, the Wigner function consists of two separate

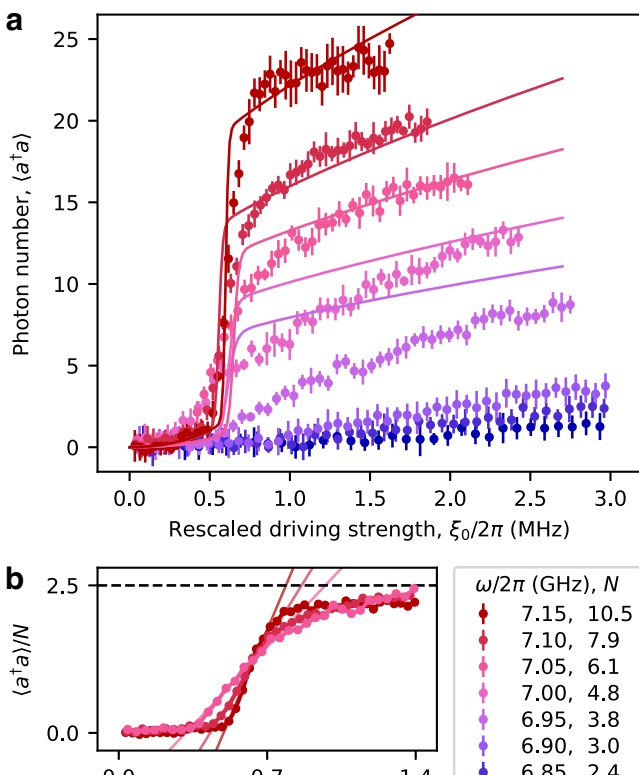

**a**

**b**

**Fig. 3 | First-order dissipative phase transition manifested by increasingly sharp photon number jump. a** When approaching the thermodynamic limit ($N \to \infty$), the observed photon number jump becomes increasingly more drastic in the classical hysteretic regime for a fixed detuning $\Delta = 3\gamma$. Here, the error bars represent the standard deviation over 8 independent experiments, and the solid lines are calculated from the quantum theory with no fitting parameter. The deviation between theory and experiment becomes increasingly large at lower resonant frequencies, which we attribute to the increasingly large dephasing rate when tuning the SQUID away from its sweet spot (See Supplementary Fig. 12). **b** Rescaled photon number $\langle a^\dagger a \rangle/N$ vs. $\xi_0/2\pi$ curves for the highest three resonant frequencies, where the dephasing effect may be fairly neglected. The solid lines show the linear fits of the transition speed, with fitted values $\partial(\langle a^\dagger a \rangle/N)/\partial(\xi_0/2\pi) = 3.8, 5.8$, and $8.5\,\text{MHz}^{-1}$, respectively. The increasingly sharp transition step indicates a first-order dissipative phase transition at $N \to \infty$. Source data are provided as a Source Data file.

parts in phase space, corresponding to the different unique SSs in the two individual phases[28,29]. The probability of staying in the coherent-state phase changes continuously into that of being in the squeezed-state phase with increasing $\xi_0$. Ideally, it reaches an equiprobable mixture of the two phases at the exact critical point, $\xi_0^*/2\pi = 0.64$ MHz[20]. One can thus understand the dip of $|\langle a \rangle|$ and the peak of $g^{(2)}(0)$ in Fig. 1c as a result of the coherent interference between the two phases. With the increase of $N$, the photon number diverges and the system behaves more classically. The SS thus must jump at $\xi_0^*$ in the thermodynamic limit, because only one potential well can be occupied at the same time in a classical system. This observation explains the increasingly sharp step of $\langle a^\dagger a \rangle/N$ with increasing $N$ in Fig. 3b, and reveals the origin of the first-order DPT.

## Discussion

The quantum behavior of the Duffing oscillator promotes the view that the extensively observed hysteresis and bistability originate from a non-classical SS around a critical point. The SS consists of two separate parts in phase space, which correspond to the two phases of the

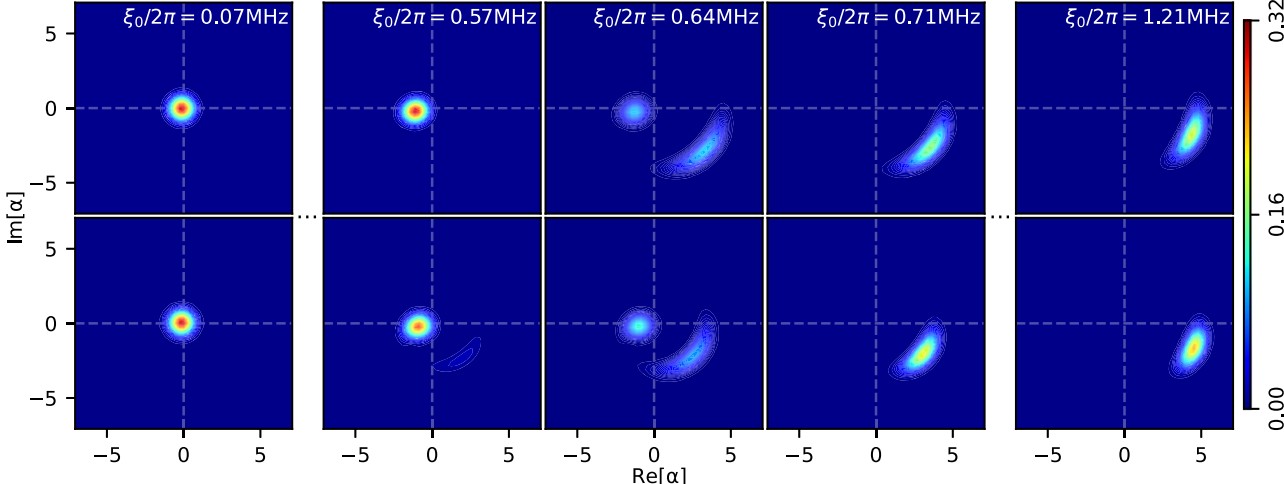

**Fig. 4 | Wigner function of the steady state during dissipative phase transition.** Shown are theory (top) with no fitting parameter and experiment (bottom) at the resonant frequency $\omega_A/2\pi = 7.15$ GHz. The steady state is approximately a coherent (squeezed) state before (after) the phase transition, which separates the two phases of the Duffing oscillator (See Supplementary Fig. 11). The transition between the two phases happens within a relatively small range, $0.57\,\text{MHz} \leq \xi_0/2\pi \leq 0.71\,\text{MHz}$, during which the steady state has two distinctive parts in the phase space and is a weighted mixture of the two phases. Ideally, it reaches an equiprobable mixture of the two phases at the exact critical point, which is around $\xi_0^*/2\pi = 0.64$ MHz. Source data are provided as a Source Data file.

system and are MSs with a remarkably long lifetime. Their lifetime diverges when approaching the thermodynamic limit and leads to a first-order DPT. Instead of seeing the intriguing dynamics as a competition between the classical and quantum tunneling rates[30], the Liouvillian spectral theory provides a simple and quantitative description for a general driven-dissipative system[18,20]. The tunable superconducting nonlinear resonator is a versatile building block for quantum simulation[31–34], and the pulsed heterodyne measurement enables the time-resolved tomography of a non-equilibrium process. We therefore expect these methods to serve as a stepping stone for simulating strongly correlated bosons in the driven-dissipative regime[35] and for unveiling the mystery of multistability from a quantum-mechanical perspective.

## Methods

### Sample preparation
The in-situ tunable Duffing oscillator consists of a 7.2 mm long necklace-type resonator with an asymmetric DC-SQUID embedded in the middle[23,24,36] (See Supplementary Fig. 1) . When applying a static magnetic field to the SQUID, we can set the resonant frequency between $\omega_A/2\pi = 5.65$ GHz and 7.15 GHz with a nonlinearity varying from $U/2\pi = -6.1$ MHz to $-58$ kHz. In this experiment, we restrict the resonant frequency in a 350 MHz range below the sweet spot, where the energy dissipation rate, $\gamma = 3.85\,\mu\text{s}^{-1}$, dominates the dephasing effect (See Supplementary Fig. 4). The sample is cooled down to a base temperature of 30 mK to suppress the thermal noise. A detailed description of the experimental setup and its characterization data can be found in Supplementary Notes 1 and 2, respectively.

### Fast flux line
For the measurements of signal moments, we apply the driving field through the flux line to avoid reflection at the RF output. The RF and DC signal of the flux line are combined by a bias-tee thermalized at the base temperature (See Supplementary Fig. 2). We describe the combined field as a collection of harmonic oscillators (bath), $H_b = \sum_{k=-\infty}^{+\infty} \omega_k b_{F,k}^\dagger b_{F,k}$, with $k$ being the wave vector and $b_{F,k}^\dagger$ ($b_{F,k}$) the creation (annihilation) operator. We assume that the system-bath interaction is a combination of a photon-preserving interaction $H_{int}^{(1)} = i\kappa_F(b_{F,k}^\dagger a - b_{F,k} a^\dagger)$ and an optomechanical-like interaction $H_{int}^{(2)} = i\kappa_\varphi(b_{F,k}^\dagger - b_{F,k})a^\dagger a$, where $\kappa_F$ and $\kappa_\varphi$ are the corresponding coupling strengths. Following the input-output analysis and neglecting the two-photon process[37], we see that $H_{int}^{(2)}$ can be neglected if the input field is close-to-resonance with the system. It indicates that a single flux line can be simultaneously used for DC bias and RF drive[38] (See Supplementary Note 1).

### Initial state preparation
When driving the system with a sufficiently weak or strong field outside the hysteretic regime, the Duffing oscillator has a single potential well that is located in proximity to either of the two wells inside the hysteretic regime. We therefore can set the driving strength to either zero or a sufficiently large value (~11 MHz in our case) and wait for approximately 4 μs for the system to reach the SS in the single potential well. Finally, we switch the driving strength to the objective value, $\xi$, in the next 250 ns that completes the initial state preparation (See Supplementary Note 3).

### Pulsed heterodyne measurement
Upon initial state preparation, we count for a waiting time, $\tau$, and measure one single period of the output signal. We repeat the procedure for $10^6$–$10^9$ times depending on the required accuracy and concatenate the data for digital filtering. We then obtain one data point of the field quadratures in each period of the concatenated signal (16 ns), and we calculate the quadrature moments to the second order and record the histogram in a 256 × 256 matrix (See Supplementary Note 3). The major advantage of the pulsed measurement setup is that it allows for the application of a narrow-band low-pass filter (cutoff frequency at 2 MHz in our case) without sacrificing the time resolution of the measurement results. The obtained 16 ns time resolution enables the observation of a non-equilibrium process.

## Data availability
The data that support the findings of this study are provided in the paper and the Supplementary Information. Source data are provided with this paper.

## Code availability
The codes for analyzing the data of this study are provided with this paper.

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

## Acknowledgements

The authors thank P. Zapletal and M. J. Hartmann for insightful discussions of dephasing effect. This work is funded by German Research Foundation via Germany's Excellence Strategy (EXC-2111-390814868), Elite Network of Bavaria through the program ExQM, European Union via the Quantum Flagship project QMiCS (No. 820505), and German Federal Ministry of Education and Research via the project QuaRaTe (No. 13N15380). This research is also part of the Munich Quantum Valley, which is supported by the Bavarian state government with funds from the Hightech Agenda Bayern Plus.

## Author contributions

Q.C. carried out theoretical calculations, performed the experiment, and analyzed data. M.F. fabricated the sample and contributed to system characterization. Y.N., M.R., and K.F. contributed to microwave techniques. E.X. and A.M. contributed to cryogenic techniques. M.P. and F.D. contributed to data analysis. S.P. contributed to FPGA programming. Q.C., F.D., and R.G. wrote the manuscript with input from all authors. R.G. supervised the project.

## Funding

## Competing interests

The authors declare no competing interests.
