## [Peer Review File · Nature Communications]

REVIEWER COMMENTS

Reviewer #1 (Remarks to the Author):

In this article, the authors experimentally demonstrate the first-order dissipative phase transition occurring in a nonlinear oscillator.

The article is extremely well written, and the authors provide an in depth study of the superconducting circuits they use, motivating their findings in great detail.

In this regard, the article is very interesting, timely, and deserves to be published. I have, however, some concerns in the way some of the results are provided, and I think answering them would improve the quality of the manuscript. I list them below. Upon answering these, I would be more than happy to support the publication of this article.

GENERAL COMMENTS

1) I think that the authors could better frame their discoveries in light of the past literature. For instance, in the abstract, the authors state: "Here, we measure the non-equilibrium dynamics of a superconducting Duffing oscillator and reconcile the classical and quantum descriptions in a unified picture of quantum metastability." This is indeed true, but other groups (theoretically Refs. 19,20,26) and experimentally (either in polaritonics, Refs. 15 and 16; or superconducting circuit but with a slightly different Hamiltonian, Ref. 17) already argued that this was the case. In this regard, I would suggest either insert to cite some of the previous works, or rephrase in order to properly acknowledge these previous works. Similarly, I would suggest to provide more perspective when discussing the difference between the classical and quantum description of the Kerr oscillator (lines 45-50) and in lines 128-129. The specific case of the Duffing oscillator was indeed theoretically investigated in great detail in Ref. 19, and the connection between the "classical" solution and the quantum observed instability was given in Ref. 20 in a general fashion. Another very interesting discussion on the connection between "classical" multi-stability and quantum "mixture" is provided in the following reference (<https://journals.aps.org/prl/abstract/10.1103/PhysRevLett.124.043601>), although the model considered is a spin system.

2) I think that the authors could briefly point out in which way their experiment differs from those in Refs. 15, 16, and 17. This could be very pedagogical for readers approaching this topic, given the broad-audience aim of the journal.

SPECIFIC COMMENTS

1) Duffing vs Kerr. As the authors do, and as it is done in much of the literature on dissipative phase transition, including the examples cited by the authors as Refs. 15, 16, and 17, the nonlinearity is called Kerr nonlinearity. And in quantum optics, the Duffing Hamiltonian is often called Kerr Hamiltonian. To this extent, I would advise the author to immediately introduce the notion that Duffing and Kerr oscillator (line 15), since these terms are often used interchangeably by the community.

2) In the abstract, the authors say "By engineering the lifetime of the metastable states sufficiently large, we observe a first-order dissipative phase transition, which mimics a sudden change of the mean field in a 11-site Bose-Hubbard lattice." This is clear, especially in view of Ref. 19. However, I think it may slightly be misleading, because the effective scaling only takes into account an effective number of sites for the uniform mode of the lattice, and not really what it is happening for a lattice of 11 resonators. To this extent, it may be also interesting to point out to the theoretical discussions in the following Refs, where finite-size effect and dimensionality is taken into account.

-- <https://journals.aps.org/pr/abstract/10.1103/PhysRevA.97.013853>

-- <https://journals.aps.org/pr/abstract/10.1103/PhysRevA.95.043826>

These works may also be used as a reference for the statement in the lines 164-166.

3) I think it may not be very clear what the authors mean by "An exceptional case occurs when the system approaches the thermodynamic limit, which mimics the mean-field model of a driven-dissipative Bose-Hubbard lattice." Furthermore, in non-Hermitian mechanics, exceptional points are a specific phenomenon, so I would avoid the term exceptional.

4) The authors state "The Liouvillian gap may even close at a critical point in the thermodynamic limit, where the eigenvalue zero has a geometric multiplicity of two but an algebraic multiplicity of one". This may be a non-standard use of algebraic and geometric multiplicity. In general, the algebraic multiplicity and geometric multiplicity of an eigenvalue can differ, but, the geometric multiplicity is always greater than the algebraic one. In the linear algebra context, the algebraic multiplicity is the number of zeros of the characteristic polynomial, while the geometric one is the dimension of the eigenspace associated with some eigenvalue. In the case of the zero eigenvalue, however, this is impossible for a Liouvillian (see, e.g., Lemma 5 of <https://journals.aps.org/pr/abstract/10.1103/PhysRevA.100.062131>).

5) I am slightly confused by Eq. (S17) in the supplementary material. To use that equation and quantify squeezing, the state needs to be Gaussian. I am not sure that at the transition point the state is Gaussian, or a superposition of two Gaussian states. Could the author provide a measure of

Gaussianity of the state? For instance, by computing the skewness and kurtosis (or other indicator of Gaussianity).

6) I think there is a typo in Eq. (S18), where the thermal dissipator should read $\mathcal{D}[\hat{a}^\dagger]$.

Reviewer #2 (Remarks to the Author):

In their manuscript, “Quantum behavior of the Duffing oscillator at the dissipative phase transition”, Chen et al. probe the out-of-equilibrium quantum dynamics in a Kerr nonlinear oscillator to settle the discrepancy between the classical and quantum description of the driven-dissipative Duffing oscillator. Classical mechanics allows two steady-states in a double well potential, giving rise to hysteretic behavior as well as bistability driven by thermal fluctuations. However, the quantum treatment yields only a single steady-state solution which obviously contradicts the classical hysteretic picture, and what is even more bizarre is the fact that two steady-states are observed even in quantum systems, where thermal fluctuations are suppressed.

In this experiment, the authors unify these two descriptions by identifying the two classical steady-states as metastable states with long lifetimes determined by the gap in the Liouvillian spectrum. Quantum hysteresis then corresponds to measuring one of the two metastable states, before the system decays to the single (true) steady state in the long time limit.

The authors perform careful time-domain measurements of the homodyne fields, photon occupation, and second-order correlations to observe the role of quantum fluctuations in a first-order DPT. The authors can probe and engineer the lifetime of the metastable states and show critical slowing down of the metastable states near the critical point - which corresponds in the thermodynamic limit to a closing of the Liouvillian gap.

On these observations alone, the paper deserves publication in Nature Communications. My comments for improving the manuscript are structural (storytelling and describing the measurement) and related to the connection to the Bose-Hubbard model (which I find a bit misleading and disruptive to the story). Once the authors resolve these points, I would be happy to recommend this manuscript for publication.

1. Initially, from reading the description in the main text it was not fully clear how the authors initialized the state only in one of the wells. I am referring to the sentence in lines 94-96: "We modulate the radio-frequency drive by three different pulse shapes...". After reading the supplemental sections S1C and S3, it became clear that this was achieved by setting the drive strength to either zero or very large value - since in that parameter regime you only have one potential well you can localize into. Fig. S7 does a better job explaining this protocol than Fig. 1A. The authors should consider expanding the explanation for the initialization step in the main text.

2. From the main text, it wasn't clear to me why the authors needed to drive the resonator through the flux line. From the supplement, the clarification was to avoid the reflecting driving field in the output path. The more confusing aspect was whether this flux drive was a coherent drive $\sim(a+a^\dagger)$ or a modulation drive of the resonator frequency $\sim(a^\dagger a)$. Experts in superconducting circuits will likely assume the latter and then realize from reading the supplement that in fact, it was the former. Ultimately this could have been done with a charge drive line - is this correct? Please clarify this in the main text.

3. For pedagogical reasons can the authors clarify for the readers (lines 123-126) why the enhanced bunching $g^2 \gg 1$ is a clear sign of a DPT.

4. The authors make a connection between the thermodynamical limit ($N \gg 1$ photons) in the zero-dimensional model (one Kerr resonator) with the thermodynamic limit in a one-dimensional driven-dissipative Bose-Hubbard model ($N \gg 1$ coupled Kerr resonators). The Bose-Hubbard model is a rich testbed for probing complex many-body dynamics which explores the vast Hilbert space under the competition of drives, dissipation, and photon interactions.

This is not the case here, the mapping is trivial as outlined in Ref. 19. A lattice of coupled oscillators has many quasi-momentum modes, tied to either the Fourier transform or diagonalizing the tight-binding Hamiltonian. However, choosing a homogeneous coherent driving field on all the cavities leads to another zero-dimensional model since you are ONLY driving the $k=0$ mode. Additionally, in this single-mode picture the photon interactions rescale as U/N (N =number of resonators) - so in the thermodynamic limit one arrives at the picture of a coherently displaced linear ($U/N \rightarrow 0$) oscillator.

This mapping of the zero-dimensional model to a one-dimensional lattice model, which is trivially a zero-dimensional model from our choice of driving scheme, is not something I find interesting or insightful, and it becomes disruptive to the main (already cool) story of the paper.

5. I find the sharp resonator response in Fig.3 a bit artificial, perhaps the authors can correct my misunderstanding. The driving strength is rescaled as $\sqrt{N} \cdot \epsilon_0$, but for all N we plot the resonator population as a function of ϵ_0 on the x-axis. If we rescale the x-axis as $\sqrt{N} \cdot \epsilon_0$

for every N , and also rescale $\langle a^\dagger a \rangle \rightarrow \langle a^\dagger a \rangle / N$, wouldn't all the curves pretty much collapse on top of each other? What am I missing?

6. In the conclusion paragraph, the authors promote lattices of nonlinear resonators as a toolbox for exploring Bose-Hubbard models with microwave photons and probing strongly-correlated quantum materials. The authors should also acknowledge the recent results from the UChicago group assembling strongly correlated quantum fluids of light in a similar architecture:

Disorder-Assisted Assembly of Strongly Correlated Fluids of Light (<https://arxiv.org/abs/2207.00577>)

NCOMMS-22-38783-T – Reply to Reviewer #1

In this article, the authors experimentally demonstrate the first-order dissipative phase transition occurring in a nonlinear oscillator. The article is extremely well written, and the authors provide an in depth study of the superconducting circuits they use, motivating their findings in great detail.

In this regard, the article is very interesting, timely, and deserves to be published. I have, however, some concerns in the way some of the results are provided, and I think answering them would improve the quality of the manuscript. I list them below. Upon answering these, I would be more than happy to support the publication of this article.

Reply: We appreciate that our work is perceived as "extremely well-written" and "in-depth" and is recommended for publication.

GENERAL COMMENTS

1) I think that the authors could better frame their discoveries in light of the past literature. For instance, in the abstract, the authors state: "Here, we measure the non-equilibrium dynamics of a superconducting Duffing oscillator and reconcile the classical and quantum descriptions in a unified picture of quantum metastability." This is indeed true, but other groups (theoretically Refs. 19, 20, 26) and experimentally (either in polaritonics, Refs. 15 and 16; or superconducting circuit but with a slightly different Hamiltonian, Ref. 17) already argued that this was the case. In this regard, I would suggest either insert to cite some of the previous works, or rephrase in order to properly acknowledge these previous works.

Reply: We rephrased this sentence as follows (line 18-20):

"Here, we measure the non-equilibrium dynamics of a superconducting Duffing oscillator and experimentally reconcile the classical and quantum descriptions as indicated by the Liouvillian spectral theory."

Besides, we rephrased the following sentence in line 56-59:

"Our experimental results settle the seeming controversy between the classical and quantum properties of the Duffing oscillator and provide support to the recent results of the Liouvillian spectral theory [18-21]"

Moreover, we inserted extra citations of these references in line 59, 130, 133, and 135.

Similarly, I would suggest to provide more perspective when discussing the difference between the classical and quantum description of the Kerr oscillator (lines 45-

50) and in lines 128-129. The specific case of the Duffing oscillator was indeed theoretically investigated in great detail in Ref.19, and the connection between the "classical" solution and the quantum observed instability was given in Ref.20 in a general fashion. Another very interesting discussion on the connection between "classical" multi-stability and quantum "mixture" is provided in the following reference (<https://journals.aps.org/prl/abstract/10.1103/PhysRevLett.124.043601>), although the model considered is a spin system.

Reply: Besides the revisions described above, we added an extra reference of the mentioned paper in the revised manuscript (Ref. 21).

2) I think that the authors could briefly point out in which way their experiment differs from those in Refs. 15, 16, and 17. This could be very pedagogical for readers approaching this topic, given the broad-audience aim of the journal.

Reply: We rephrased the following sentence in line 47-56:

"Recently, signatures of a dissipative phase transition (DPT) have been observed in the scattering coefficient [14,15], decay rate [15,16], and second-order correlation function [17] of the Duffing oscillator, which indicate a prominent role of quantum fluctuation in the SS. These experiments are performed around a fixed parameter setting using a continuous-wave measurement setup.

Here, we simulate the non-equilibrium quantum dynamics of the Duffing oscillator with an *in-situ* tunable superconducting nonlinear resonator. Besides the wide tunability range of sample parameters in one device, the pulsed heterodyne measurement distinguishes our experiment from the experiments already reported in the literature. Our experimental setup allows for a proper control of the initial state at different parameter settings as well as a high time-resolution readout."

SPECIFIC COMMENTS

1) Duffing vs Kerr. As the authors do, and as it is done in much of the literature on dissipative phase transition, including the examples cited by the authors as Refs. 15, 16, and 17, the nonlinearity is called Kerr nonlinearity. And in quantum optics, the Duffing Hamiltonian is often called Kerr Hamiltonian. To this extent, I would advise the author to immediately introduce the notion that Duffing and Kerr oscillator (line 15), since these terms are often used interchangeably by the community.

Reply: We emphasized "Kerr nonlinearity" in line 32, and we formally defined the system Hamiltonian as "Kerr-oscillator Hamiltonian with coherent drive" (line 72).

2) In the abstract, the authors say "By engineering the lifetime of the metastable states sufficiently large, we observe a first-order dissipative phase transition, which mimics a sudden change of the mean field in a 11-site Bose-Hubbard lattice." This is clear, especially in view of Ref. 19. However, I think it may slightly be misleading, because the effective scaling only takes into account an effective number of sites for the uniform mode of the lattice, and not really what it is happening for a lattice of 11 resonators. To this extent, it may be also interesting to point out to the theoretical discussions in the following Refs, where finite-size effect and dimensionality is taken into account.

– <https://journals.aps.org/pr/abstract/10.1103/PhysRevA.97.013853>

– <https://journals.aps.org/pr/abstract/10.1103/PhysRevA.95.043826>

These works may also be used as a reference for the statement in the lines 164-166.

Reply: As pointed by both referees that the discussion of Bose-Hubbard lattice is disruptive to our main story, we removed the unnecessary discussions except for introducing the thermodynamic limit (line 168-172).

3) I think it may not be very clear what the authors mean by "An exceptional case occurs when the system approaches the thermodynamic limit, which mimics the mean-field model of a driven-dissipative Bose-Hubbard lattice." Furthermore, in non-Hermitian mechanics, exceptional points are a specific phenomenon, so I would avoid the term exceptional.

Reply: We rephrased this sentence as follows (line 63-65):

"A remarkable case occurs when the system approaches the thermodynamic limit where the MSs gain an increasingly long lifetime when approaching a critical point but, suddenly, can no longer be properly defined at the exact point [19,20]."

4) The authors state "The Liouvillian gap may even close at a critical point in the thermodynamic limit, where the eigenvalue zero has a geometric multiplicity of two but an algebraic multiplicity of one". This may be a non-standard use of algebraic and geometric multiplicity. In general, the algebraic multiplicity and geometric multiplicity of an eigenvalue can differ, but, the geometric multiplicity is always greater than the algebraic one. In the linear algebra context, the algebraic multiplicity is the number of zeros of the characteristic polynomial, while the geometric one is the dimension of the eigenspace associated with some eigenvalue. In the case of the zero eigenvalue, however, this is impossible for a Liouvillian (see, e.g., Lemma 5 of <https://journals.aps.org/pr/abstract/10.1103/PhysRevA.100.062131>).

Reply: We thank the referee for pointing out the misuse of the mathematical notations. We rephrased this sentence as follows (line 89-90):

”The Liouvillian gap may even close at a single critical point in the thermodynamic limit, where the eigenvalue *zero* becomes degenerate at the exact point.”

5) I am slightly confused by Eq. (S17) in the supplementary material. To use that equation and quantify squeezing, the state needs to be Gaussian. I am not sure that at the transition point the state is Gaussian, or a superposition of two Gaussian states. Could the author provide a measure of Gaussianity of the state? For instance, by computing the skewness and kurtosis (or other indicator of Gaussianity).

Reply: The referee is correct that the Gaussian-state approximation is not valid during the phase transition process. However, the approximation should be fairly good outside this small hysteretic regime, and thus the squeezing level can be used to distinguish the two different phases [see J. Opt. B: Quantum Semiclass. 6, 387 (2004)].

As suggested by the referee, we provided a measure of Gaussianity in Fig. S10 B-C (also attached here). We showed that the 3rd order cumulants, $\langle\langle a^{\dagger 3} \rangle\rangle$ and $\langle\langle a^{\dagger 2} a \rangle\rangle$, are fairly small compared to the corresponding 3rd order moments, $\langle a^{\dagger 3} \rangle$ and $\langle a^{\dagger 2} a \rangle$, outside the hysteretic regime. The experimental data shows a good consistency with the theoretical curves.

6) I think there is a typo in Eq. (S18), where the thermal dissipator should read $\mathcal{D}[\hat{a}^\dagger]$.

Reply: We thank the referee for pointing out this typo, which has been corrected in the revised manuscript.

NCOMMS-22-38783-T – Reply to Reviewer #2

In their manuscript, "Quantum behavior of the Duffing oscillator at the dissipative phase transition", Chen et al. probe the out-of-equilibrium quantum dynamics in a Kerr nonlinear oscillator to settle the discrepancy between the classical and quantum description of the driven-dissipative Duffing oscillator. Classical mechanics allows two steady-states in a double well potential, giving rise to hysteretic behavior as well as bistability driven by thermal fluctuations. However, the quantum treatment yields only a single steady-state solution which obviously contradicts the classical hysteretic picture, and what is even more bizarre is the fact that two steady-states are observed even in quantum systems, where thermal fluctuations are suppressed.

In this experiment, the authors unify these two descriptions by identifying the two classical steady-states as metastable states with long lifetimes determined by the gap in the Liouvillian spectrum. Quantum hysteresis then corresponds to measuring one of the two metastable states, before the system decays to the single (true) steady state in the long time limit.

The authors perform careful time-domain measurements of the homodyne fields, photon occupation, and second-order correlations to observe the role of quantum fluctuations in a first-order DPT. The authors can probe and engineer the lifetime of the metastable states and show critical slowing down of the metastable states near the critical point - which corresponds in the thermodynamic limit to a closing of the Liouvillian gap.

On these observations alone, the paper deserves publication in Nature Communications. My comments for improving the manuscript are structural (storytelling and describing the measurement) and related to the connection to the Bose-Hubbard model (which I find a bit misleading and disruptive to the story). Once the authors resolve these points, I would be happy to recommend this manuscript for publication.

Reply: We thank the referee for appreciating both the scientific and technological interests of our work as well as the recommendation for publication.

1. Initially, from reading the description in the main text it was not fully clear how the authors initialized the state only in one of the wells. I am referring to the sentence in lines 94-96: "We modulate the radio-frequency drive by three different pulse shapes...". After reading the supplemental sections S1C and S3, it became clear that this was achieved by setting the drive strength to either zero or very large value - since in that parameter regime you only have one potential well you can localize into. Fig. S7 does a better job explaining this protocol than Fig. 1A. The authors should consider expanding the explanation for the

initialization step in the main text.

Reply: We added a brief description of initial state preparation in the Methods section.

Besides, we rephrased the following sentence in line 99-101:

”We modulate the radio-frequency (RF) drive by three different pulse shapes, which balance the depths of the two potential wells and prepare the system in one of the two wells or in the SS at the initial time (see Methods section).”

2. From the main text, it wasn't clear to me why the authors needed to drive the resonator through the flux line. From the supplement, the clarification was to avoid the reflecting driving field in the output path. The more confusing aspect was whether this flux drive was a coherent drive $\sim (a + a^\dagger)$ or a modulation drive of the resonator frequency $\sim (a^\dagger a)$. Experts in superconducting circuits will likely assume the latter and then realize from reading the supplement that in fact, it was the former. Ultimately this could have been done with a charge drive line - is this correct? Please clarify this in the main text.

Reply: The referee is correct that an extra charge line would be a more straightforward choice. The practical reason of driving through the flux line is that we have underestimated the influence of the reflecting field in the sample design. There is no separate charge line for driving. However, the input-output analysis indicates that driving through the flux line plays the same role. This method is essentially the same with the XYZ-line reported in *Appl. Phys. Lett.* 119, 144001 (2021). It leads to a practical advantage that we can save one port for each resonator when scaling up the system, and is beneficial to our future experiments.

We added a brief explanation of the fast flux line in the Methods section.

3. For pedagogical reasons can the authors clarify for the readers (lines 123-126) why the enhanced bunching $g^2 \gg 1$ is a clear sign of a DPT.

Reply: The SSs around the critical point of the 1st-order DPT has a drastic change. Let us label them as low-photon and high-photon states, ρ_L and ρ_H , respectively. The steady states at the critical point is therefore $\rho_{SS} = (\rho_L + \rho_H)/2$. Photon bunching happens if $\text{Tr}(a^\dagger a \rho_{SS'}) \geq \text{Tr}(a^\dagger a \rho_{SS})$, where $\rho_{SS'} = a \rho_{SS} a^\dagger$ is the state after one-photon subtraction. One can see that the probability of ρ_H in $\rho_{SS'}$ is larger than 1/2 because of the back-action of photon subtraction, and we therefore have $g^2 \gg 1$ [see also *Nat. Phys.* 14, 365-369 (2017)].

We rephrased the following sentence in line 129-130:

”This is a typical signature of a first-order DPT, resulting from a drastic change of the SSs on the two sides of a single critical point [17].”

4. The authors make a connection between the thermodynamical limit ($N \gg 1$ photons) in the zero-dimensional model (one Kerr resonator) with the thermodynamic limit in a one-dimensional driven-dissipative Bose-Hubbard model ($N \gg 1$ coupled Kerr resonators). The Bose-Hubbard model is a rich testbed for probing complex many-body dynamics which explores the vast Hilbert space under the competition of drives, dissipation, and photon interactions.

This is not the case here, the mapping is trivial as outlined in Ref. 19. A lattice of coupled oscillators has many quasi-momentum modes, tied to either the Fourier transform or diagonalizing the tight-binding Hamiltonian. However, choosing a homogeneous coherent driving field on all the cavities leads to another zero-dimensional model since you are ONLY driving the $k = 0$ mode. Additionally, in this single-mode picture the photon interactions rescale as U/N ($N =$ number of resonators) - so in the thermodynamic limit one arrives at the picture of a coherently displaced linear ($U/N \rightarrow 0$) oscillator.

This mapping of the zero-dimensional model to a one-dimensional lattice model, which is trivially a zero-dimensional model from our choice of driving scheme, is not something I find interesting or insightful, and it becomes disruptive to the main (already cool) story of the paper.

Reply: We agree that the discussion of Bose-Hubbard lattice is disruptive to our main story. We removed the unnecessary discussions except for introducing the thermodynamic limit (line 168-172).

5. I find the sharp resonator response in Fig. 3 a bit artificial, perhaps the authors can correct my misunderstanding. The driving strength is rescaled as $\sqrt{N}\epsilon_0$, but for all N we plot the resonator population as a function of ϵ_0 on the x -axis. If we rescale the x -axis as $\sqrt{(N)}\epsilon_0$ for every N , and also rescale \rightarrow /N , wouldn't all the curves pretty much collapse on top of each other? What am I missing?

Reply: Here, we plot the raw data and the rescaled data as the referee described.

The referee is correct that ”all the curves pretty much collapse on top of each other”. This is because the tuning range of N varies only by a factor of two. Those low frequency curves ($\omega_A/2\pi \leq 7$ GHz) are largely influenced by dephasing and are thus not suitable for a fair comparison. However, one can still observe a subtle difference between the three high-frequency curves, where the switching speed gets increasingly large with N . This trend is visible by more than one standard variation.

We note that the three resonance frequencies for simulation are fine-tuned from 7.1497, 7.1008, 7.0501 GHz to 7.1493, 7.1005, 7.0497 GHz. The power gain of the measurement setup is manually adjusted by -0.58 dB for all the curves. We did not adjust the driving strength (offset voltage, input attenuation, etc.) which will lead to a better fit between the simulation and the experiment.

6. In the conclusion paragraph, the authors promote lattices of nonlinear resonators as a toolbox for exploring Bose-Hubbard models with microwave photons and probing strongly-correlated quantum materials. The authors should also acknowledge the recent results from the UChicago group assembling strongly correlated quantum fluids of light in a similar architecture: Disorder-Assisted Assembly of Strongly Correlated Fluids of Light (<https://arxiv.org/abs/2207.00577>)

Reply: This reference is highly relevant to the outlook of our work, and we cited it as Ref. 34 in the revised manuscript.

REVIEWERS' COMMENTS

Reviewer #1 (Remarks to the Author):

The authors replied in a very convincing way to all the remarks and implemented all the requested changes.

I thus maintain my point that this work deserves publication in Nature Communications, and I think that in the current version the manuscript meets all the requirements for publication.

Reviewer #2 (Remarks to the Author):

The authors have resolved most of my comments, and have improved the story and structure of the manuscript.

I thank the authors for plotting the rescaled data of Fig.3. My initial concern with this figure was that the resonator response vs N looked sharper than it actually is. The authors acknowledged that the curves pretty much collapse on top of each other when rescaling the data and drive axis. That is to be expected given the small tuning range for the photon number N. I agree with the authors, there is a difference in the switching speed for the three high-frequency curves -- but it is subtle and not as dramatic as the authors describe in lines 202-203: "This observation explains the increasingly sharp step of $\langle a^\dagger a \rangle$ with increasing N in Fig. 3, and reveals the origin of the first-order DPT."

For transparency, I still consider it important to show the rescaled dataset with accompanying clarification regarding the switching speeds. The authors can either attach this to Fig.3 in the main text or add it to the methods/supplement and reference it in the main text.

After the authors resolve this last comment, I would be happy to recommend this manuscript for publication.

NCOMMS-22-38783A – Reply to Reviewer #1

The authors replied in a very convincing way to all the remarks and implemented all the requested changes.

I thus maintain my point that this work deserves publication in Nature Communications, and I think that in the current version the manuscript meets all the requirements for publication.

Reply: We thank the referee for the recommendation of publication.

NCOMMS-22-38783A – Reply to Reviewer #2

The authors have resolved most of my comments, and have improved the story and structure of the manuscript.

I thank the authors for plotting the rescaled data of Fig.3. My initial concern with this figure was that the resonator response vs N looked sharper than it actually is. The authors acknowledged that the curves pretty much collapse on top of each other when rescaling the data and drive axis. That is to be expected given the small tuning range for the photon number N. I agree with the authors, there is a difference in the switching speed for the three high-frequency curves – but it is subtle and not as dramatic as the authors describe in lines 202-203: "This observation explains the increasingly sharp step of with increasing N in Fig. 3, and reveals the origin of the first-order DPT."

For transparency, I still consider it important to show the rescaled dataset with accompanying clarification regarding the switching speeds. The authors can either attach this to Fig.3 in the main text or add it to the methods/supplement and reference it in the main text.

After the authors resolve this last comment, I would be happy to recommend this manuscript for publication.

Reply: We thank the referee for the recommendation of publication after revising Fig. 3.

We plotted the rescaled data in Fig. 3b as suggested by the referee, which was previously provided in our reply. We fitted the switching rates by linear functions and clarified the observation in both the figure legend and the lines 172-180.